# Cost-Effectiveness of Glucosamine in Osteoarthritis Treatment: A Systematic Review

**DOI:** 10.3390/healthcare11162340

**Published:** 2023-08-18

**Authors:** Nam Xuan Vo, Ngan Nguyen Hoang Le, Trinh Dang Phuong Chu, Huong Lai Pham, Khang Xuan An Dinh, Uyen Thi Thuc Che, Thanh Thi Thanh Ngo, Tien Thuy Bui

**Affiliations:** 1Faculty of Pharmacy, Ton Duc Thang University, Ho Chi Minh City 700000, Vietnam; hoangngan25080177@gmail.com (N.N.H.L.); trinhchu140901@gmail.com (T.D.P.C.); laihuong49@gmail.com (H.L.P.); dinhxuanankhang94@gmail.com (K.X.A.D.); thucuyenpct@gmail.com (U.T.T.C.); thahthah0906@gmail.com (T.T.T.N.); 2Faculty of Pharmacy, Le Van Thinh Hospital, Ho Chi Minh City 700000, Vietnam; thuytienbui2404@gmail.com

**Keywords:** economic evaluation, cost-effectiveness, glucosamine, osteoarthritis, systematic review

## Abstract

Osteoarthritis (OA) is a chronic condition that most frequently affects older adults. It is currently the most common disability. The cost of treating an aging population places pressure on the healthcare budget. As a result, it is imperative to evaluate medicines’ cost-effectiveness and, accordingly, their influence on health resource allocation. Our study aims to summarize the cost and outcome of utilizing glucosamine in OA treatment. Databases like Medline, Cochrane, and Scopus were searched as part of the identification process up until April 2023. Our primary inclusion criteria centered on the economic evaluation of glucosamine in OA treatments, providing an incremental cost-effectiveness ratio (ICER). The Quality of Health Economic Studies (QHES) instrument was applied to grade the quality of the studies. Seven qualified studies that discussed the cost-effectiveness of glucosamine with or without other formulations were selected. All of them demonstrated that glucosamine was cost-effective. There was an increase in quality-adjusted life years (QALYs) when incorporating glucosamine in conventional care. Moreover, patented crystalline glucosamine sulfate (pCGS) was more cost-effective than the other formulations of glucosamine (OFG). Overall, utilizing pCGS was more beneficial than using OFG in terms both of cost and quality of life.

## 1. Introduction

Due to its growing prevalence, osteoarthritis (OA) poses a significant challenge to healthcare budgets. OA is a chronic illness characterized by the degradation of cartilage in joints, which causes bones to rub against one another; this ultimately leads to pain, stiffness, edema, and disability, with a detrimental impact on patients’ quality of life (i.e., all patients with OA) [1,2,3,4]. The current global prevalence of osteoarthritis (OA) is greater than 7% of the population (528 million people), reaching up to 14% in countries with aging populations and established market economies [3,5]. The global prevalence of OA increased by 48% from 1990 to 2019 in different geographical regions. This rate will continue to increase in regions with aging populations and established market economies, such as Europe and North America [5,6,7,8,9,10,11]. For example, the prevalence of osteoarthritis of the knee and hip is highest in North America (5924 per 100,000 individuals), followed by North Africa, the Middle East (4610 per 100,000), and Australia (4595 per 100,000) [5,6,12]. In contrast, the rates of osteoarthritis are much lower in Eastern Sub-Saharan Africa (2568 per 100,000), Central Sub-Saharan Africa (2633 per 100,000), and Western Sub-Saharan Africa (2678 per 100,000) [5,6,12].

Osteoarthritis is the 15th leading cause of disability, accounting for 2.2% of 43 total global years of disability (YLD) (18.9 million in 2019) [5]. Although osteoarthritis can occur in any joint in the body, it most frequently occurs in the knee joint, accounting for 365 million cases worldwide and 61% of YLDs lost to knee osteoarthritis, followed by hand osteoarthritis (142 million cases and 24% YLD osteoarthritis) and hip osteoarthritis (33 million and 5.5% of OA YLDs) [5,6,13,14]. As demonstrated by the Vietnam Musculoskeletal Association, the prevalence of arthritis in people over 35 years old is about 30%, while in people over 65 years old, it is about 60%, and it reaches 85% in people over 80 years old [15]. Based on a study conducted in 2003, the proportion of musculoskeletal pain in the urban population in Vietnam was 14.5% and OA was the most common form of arthritis [16].

The number of people with OA continues to increase. It now not only affects the elderly, but also many young people. Risk factors for OA include age, obesity, sex, malformations, previous joint damage, or employment with a high risk of joint injury [1,7,8,9,10,11,12,13,14,17,18]. The more risk factors a person has, the more likely they are to develop osteoarthritis. Obesity in particular is a potential risk factor for developing OA [19]. For example, obesity triples the risk of knee osteoarthritis [18]. In addition, 35% of men and 62% of women have reported experiencing knee pain [16]. According to statistics published by the Dutch Institute for Public Health, the prevalence of knee OA in people 55 and older was 15.6% for males and 30.5% for females [1]. These results demonstrated that women were more likely to suffer from OA than men. Additionally, another study showed that the range for the prevalence of OA is 20.5% to 68.0%, and in several Asian countries, the majority of the Asian inhabitants reported having knee OA in a range from 13.1% to 71.1% [8,20]. Risk variables such as age, sex (specifically female), and obesity have been linked to OA. Other significant risk factors for OA include osteoporosis, higher body mass density, muscle function, ethnicity and race, genetics, low levels of education, family history, smoking, lifestyle, and environmental variables [17,19,20].

OA is a significant public health issue with few viable medical treatments [1,2,3,4,13,21,22]. Moreover, it also influences countries’ mortality rates, prevalence rates, and medical costs [3,4,13,22]. According to records from 1995, OA reportedly affected more than 1.2 million Australians and caused severe damage to quality of life and expenses [23]. The cost was determined to be up to 1090 million dollars up until 2001 [24]. Globally, osteoarthritis is present in 22.9% of adults over the age of 40 [25]. The number of people with OA is gradually rising as the global population ages [15]. Thus, the economic burden of OA, which comprises both direct and indirect medical costs, is believed to be substantial. The direct costs of osteoarthritis treatment can be as high as 1–2.5% of gross national product in countries such as the US, UK, Canada, and Australia [26,27]. Many researchers have evaluated the degree of economic burden caused by osteoarthritis; for instance, a review by Salmon et al. in 2016 addressed the economic impacts of lower-extremity (knee and hip) degenerative joint disease, both in terms of direct and indirect health costs, in different countries from the perspective of taxpayers and society as a whole [28].

In France, the direct cost of OA was estimated to be EUR 1.64 billion in 2001 [29], while 69.9 million people were reported to have OA in the United States in the same year [30,31]. The direct medical costs of osteoarthritis treatment in the United States are estimated at USD 72 billion (using median cost data from 2008 to 2011) [32]. In 2013, osteoarthritis was the second most expensive medical condition of all diseases treated in US hospitals, accounting for 4.3% (USD 18.4 billion) in total hospitalization costs (USD 415 billion) [33]. Although healthcare costs are much greater in the United States than those in other high-income countries, the direct costs of osteoarthritis treatment in those other countries remain substantial [34]. In Australia, for example, direct medical costs for osteoarthritis were an estimated AUD 1.7 billion in 2015, about 2.4% of the cost of treating arthritis in the United States, despite the population being approximately 7.3% of the size of the United States population (2015). The indirect costs of treating osteoarthritis are also significant. Published estimates of the indirect costs of osteoarthritis in different established market economies include Spain (USD 1.2 billion), the United Kingdom (USD 6.5 billion), and the United States (USD 12.7 billion) [35,36].

To reduce symptoms and enhance patients’ quality of life, numerous scientific organizations have presented therapeutic options for osteoarthritis, including pharmacological and non-pharmacological therapies [37,38]. In accordance with the European Society for Clinical and Economic Aspects of Osteoporosis, Osteoarthritis, and Musculoskeletal Diseases (ESCEO) recommendations, it is advisable to utilize symptomatic slow-acting drugs (SYSADOAs) from the beginning of OA pharmacological treatment [39]. SYSADOA groups include many other compounds, such as glucosamine, chondroitin, diacerein, and avocado soybean (unsaponifiable) [39]. For nearly 40 years, glucosamine and chondroitin sulfate (CS), two components of articular cartilage’s extracellular matrix, have been utilized medicinally [40]. These substances are widely used in formulations of both pharmaceutical products and cosmetics. It should be emphasized that not all of these substances have been clinically proven to be beneficial, though they are believed to have therapeutic effects [39]. In the process of investigating glucosamine products, ESCEO highlights that only patented crystalline glucosamine sulfate (pCGS) should be used for prescription-level medications; other formulations of glucosamine are not recommended [39]. Glucosamine sulfate is one alternative solution used to treat mild-to-moderate OA patients [15]. The assessment of glucosamine’s cost-effectiveness based on scientific studies has mainly concentrated on comparing cost-effectiveness among various formulations or with other therapies, whereas, overall, reviews exploring the financial efficacy of glucosamine are very limited. In addition, state management agencies relied on pharmacoeconomic-related data to determine the type of resource allocation that would produce the greatest efficacy; hence, these evaluations are crucial for setting price limits and reimbursement [41]. Furthermore, since we cannot access patients’ personal data due to technical and legal issues, as well as patient consent, we conducted our assessment based on scientific data.

In the absence of published scientific evidence evaluating the cost-effectiveness of glucosamine, researchers are heavily reliant on published research papers and unpublished presentations provided by academic or field-related researchers. The primary goal of this study is to evaluate the economic efficacy of glucosamine in the real world for the treatment of osteoarthritis and to summarize the main findings.

## 2. Materials and Methods

### 2.1. Search Strategy

The searching process started on 25 October 2022. We searched databases, such as Medline (using PubMed), Scopus (using www.scopus.com (accessed on 14 January 2023)), and Cochrance (using www.cochrance.org (accessed on 17 January 2023)), to find results about health-related quality of life published after April 2023. Three databases were chosen based on convenience, which were free and easy to access in a realistic situation. Our study aimed to identify all relevant articles that provided detailed information about glucosamine’s economic evaluation in order to automatically compare means with one another.

The investigation strategy involved using specific keywords for systematic review, searching phrases for measurement features, and a verified methodological search filter for measurement properties [42]. The references section of each manuscript was also examined to identify further relevant studies. Other techniques were also used in the search process, such as connecting keywords using a Boolean formula (AND, OR), selecting the specified field (title, author, summary, year, or all fields, …), checking duplicates, and reviewing articles’ conformity with inclusion criteria. The complete syntax used in this study was: (((((Rheumatoid arthritis) OR (RA)) OR ((osteoarthritis) OR (OA))) AND (glucosamin*))) AND (cost [MeSH Terms]). For the Scopus database search, the syntax was modified to the same meaning as that used for the PubMed and Cochrane databases.

### 2.2. Selection Criteria

The inclusion criteria focused on publications that met the following requirements:The interventions discussed used glucosamine as the non-combined formulation;The paper was written in English and either evaluated the cost-effectiveness or contained any other type of economic evaluation;The main topic of the paper was osteoarthritis therapy with a viable duration;The paper contained specific information reporting the ICER value;The research presented a clear conclusion as to whether glucosamine was cost-effective or not.

The exclusion criteria were as follows:The study combined glucosamine with other compounds;The study was not available to read in the English language;The study did not discuss osteoarthritis treatment, or did not focus on glucosamine;The study did not relate to OA treatments;The study contained an unclear statement or lacked information about the ICER.

### 2.3. Data Extraction

Our study placed the most emphasis on the incremental cost-effectiveness ratio (ICER) as a primary outcome measure. The ICER is a specific value that can be expressed as the price for each quality-adjusted life-year (QALY) gained [43]. The QALY assumes that a year of life lived in perfect health is worth 1 QALY [44]. Additional information, such as the first author and year of publication, OA subjects, intervention, country, perspective, type of model and tool used, the time frame of the study, and the main assumptions, were also extracted as high-yield data during the examination. Figure 1 shows the entire search process based on abstract and inclusion criteria.

### 2.4. Quality Assessment of Selected Articles

To evaluate the quality of the included studies, we employed the Quality of Health Economic Studies (QHES) instrument. The 16 questions in this application were developed in order to examine adequate methodologies, reliable data, and thorough findings in each CEA report [45]. By summing up all the points for “yes” answers to the questions, the quality score was obtained [45]. The score varied from 0 to 100. Reports with a total score of <75 were deemed to be of “low” quality, while those with a final value of >75 could qualify as being “high” quality.

## 3. Results

### 3.1. Study Selection Process

Using the keywords “rheumatoid arthritis”, “osteoarthritis”, “glucosamine”, and “cost”, we gathered articles from electronic databases while adhering to the established syntax. As a result, 63 articles from PubMed, 35 articles from Cochrane, and 147 articles from Scopus were identified. We eliminated any duplicates, resulting in a shortlist of 94 manuscripts. By applying our criteria, we removed 83 publications (due to insufficient information about expenses) and a further 4 manuscripts that did not include glucosamine. Ultimately, seven studies met the requirements for inclusion. Figure 1 depicts an overview of the selection procedure.

### 3.2. Characteristics of Included Studies

As summarized in Table 1, the reports tended to examine distinct kinds of glucosamine (pCGS or other forms of glucosamine) or to compare different therapies, such as analgesics, whereas the financial assessment took the minority position. Additionally, studies were published between 2004 and 2023. Our subjects included patients diagnosed with Osteoarthritis; two of the studies were knee-focused.

Table 2a,b displays the primary characteristics of the included studies. Noting that aging was a major risk factor for osteoarthritis, it is important to be aware that the subjects in the published studies were mostly OA patients over 40 years of age. The majority of the studies were conducted in industrialized nations, such as the UK, Germany, and Spain. The sole study conducted in Southeast Asia was based in Thailand.

On the other hand, only four of the seven publications in our research—which included societal, healthcare, and national healthcare systems—mentioned the perspective of taxpayers. In addition, a variety of models were applied to determine or convert to utility scores. According to research by Bruyère et al. [46,47,48], the utility score and QALYs were estimated from the data of published clinical trials using Grootendorst’s linear regression model based on WOMAC, the basic demographics, and the severity of OA. The Markov model or a decision tree mathematical model were employed in the other investigations’ approaches. Except for Segal et al.’s study, which adopted cost–utility analysis (CUA), the majority of research papers we chose to highlight used cost-effectiveness analysis (CEA). It should be noted that only 3/7 studies presented information about the time horizon. While Scholtissen et al. [41] reported a 6-month time horizon, Luksameesate et al. [15] and Black et al. [49] both documented lifetime horizons.

Moreover, the length of the studies varied, with some lasting at least 6 months and at most 3 years. The ICER was extracted from most of the papers as the main outcome. Sensitivity analysis was also performed in certain studies. Sensitivity analyses had been performed in some studies, in which the researchers found utility and discount rates to elicit the biggest effects.

The studies evaluated took place over a long period of time and over different time periods, so they needed to be adjusted to the same time period and the same units of comparison. Therefore, applying the appropriate discount rates was critical to bringing the values back to the present time. The discount rate ranged from 3% to 5%. The discount rate specified in Luksameesate et al.’s [15] study was 3%, as recommended by the Guideline for Health Technology Assessment. The discount rates in the Segal et al. [50] study and Black et al. [49] study were 3.5% and 5%, respectively.

### 3.3. Quality Assessment by QHES Instrument

As described in Appendix A, the QHES scores in seven studies on glucosamine in OA treatment ranged from 88 to 95, with an average of 90.6. All seven studies scored above 75, implying that these can be considered high-quality studies. These articles also clearly described unit costs and outcomes, and incremental analysis was performed between alternatives for resources and costs. Only 14% of these studies included the perspective of the analysis, while approximately 29% provided justification for the discount rate used. All of them had a statement disclosing the source of funding for the study.

### 3.4. Keypoint Data Related to Cost-Effectiveness

As shown in Table 3, the studies by Bruyère et al. [46,47,48] in 2019, 2021, and 2023 reported a slight increase in QALYs over periods of 3 months, 6 months, and 36 months. In Bruyère et al.’s 2021 study [47], the costs of pCGS over 3 months, 6 months, and 36 months were EUR 77.0964, EUR 183.0003, and EUR 2785.2712, respectively. In Bruyère et al.’s 2023 [46] study, the ICERs of pCGS and OFG after 3 months of use were 3165 USD/QALY and 32,400 USD/QALY; after 6 months, that of pCGS was 3069 USD/QALY; meanwhile, the placebo was better than OFG. The result shows that the use of pCGS is economically viable compared with the threshold of USD 3260/QALY, while OFG is not economically efficient compared with the threshold. All three of Bruyère et al.’s [46,47,48] studies suggested that pCGS was more cost-effective than OFG. The early addition of pCGS to the standard care increased QALY by 0.87. Adding pCGS increased efficiency and thus saved money. In Scholtissen’s study [41], the ICERs for glucosamine compared with paracetamol and placebo were 1376 EUR/QALY and 3617.47 EUR/QALY, respectively. When compared with the threshold of EUR 20,000/QALY, it was found to be more economically viable than using paracetamol. In the study by Segal et al. [50], after using glucosamine, the QALY gain was 0.052 and cost was USD 180.024. Glucosamine was more economical than nonsteroidal anti-inflammatory drugs (NSAIDs), with ICER of USD 3462/QALY. Adding glucosamine to existing care cost GBP 2346.85. It has also been shown to be economically viable, with an ICER of 21,33 GBP/QALY compared with the willingness-to-pay threshold of 22,000 GBP/QALY.
ICER = ΔCost/ΔQALY (incremental costs/incremental QALY gained)

## 4. Discussion

This study sought to assess the cost-effectiveness of glucosamine in the treatment of OA. To delay disease progression and control symptoms efficiently, patients could access pharmacological treatments, such as paracetamol, NSAIDs, SYSADOA, and intra-articular corticosteroids, or a surgical option, such as total knee arthroplasty (TKA). Statistical approaches for CEA have been developed, and a measure known as the incremental cost-effectiveness ratio (ICER) has gained widespread acceptance among researchers and governments [51]. The ICER is calculated by dividing the cost difference between two strategies by the difference in efficacy. This one-dimensional summary metric assesses the trade-offs between patient outcomes gained and resources spent. It can be defined as the cost of acquiring one extra unit of efficacy. The ICER threshold can be interpreted as the maximum amount society is willing to spend for an additional unit of healthcare benefit [52]. Among those medications, glucosamine was proven to slow down disease progression [53,54] and to be cost-effective at 5000 AUD/QALY [24]. In addition, our main findings showed that interventions utilizing patented pCGS were more cost-effective than those using alternative glucosamine formulations. Glucosamine sulfate was found to be more cost-effective than paracetamol [41]. Several cost-effectiveness analyses have been published from 2019 to 2023. Studies can vary, comparing cost-effectiveness between forms of glucosamine, paracetamol, and NSAIDs, or combining glucosamine with other interventions. This shows the potential of using glucosamine in OA patients. Conducting research from a healthcare perspective helps governments to evaluate and incorporate glucosamine into government policies.

Most papers required an estimation of cost–utility to compute QALYs, since there was no direct assessment of cost–utility value or QALYs; hence, costs would also be derived from accessible data sources. Following Grootendorst’s formulation, which was based on age, the number of years since an OA diagnosis, and three separate WOMAC sub-scores (Western Ontario and McMaster University Osteoarthritis Index), the health-related cost–utility was calculated. Then, the ICER was computed and reviewed. For instance, investigations in 2019 and 2021 found that the application of glucosamine sulfate was much more economical than that of the other formulations after assessing 10 articles using the WOMAC scale (4 studies using pCGS and 6 using non-pCGS formulations) [47,48]. The average ICERs of 3 months, 6 months, and 3 years were calculated to be 4489 EUR/QALY; 5347.2 EUR/QALY; 9983 EUR/QALY, respectively [47].

The time horizons were poorly reported in the studies we assessed. According to one report evaluating the role of the time horizon in CEA, the author concluded that the ICER was strongly dependent on the time horizon used [55]. Its findings revealed that most CEAs were more cost-effective as the time horizon increased [55]. Since osteoarthritis is a chronic condition, a lifetime horizon should be considered to ensure that all costs and benefits are adequately captured. Sensitivity analysis was included in CEA to find values that affected the results. It also demonstrated their reliability. Sensitivity analysis was used to check the results of the model when changing one of the parameters. In Luksamees et al.’s [15] study, they included parameters such as the cost of crystalline glucosamine sulfate, the cost of TKA, the transition probability of diclofenac plus proton pump inhibitor (PPI), the transition probability of TA injection, the transition probability of TKA, and the utility of knee OA pain. They found that the highest impact resulted from the utility of moderate pain when changing the values between 0.35 and 0.77. The cost of TKA was more sensitive than the cost-effectiveness ratio when the cost-effectiveness ratio changed between THB 78,533 and THB 79,316.

According to Scholtissen et al. [41], the average total cost per patient that used glucosamine was 38.88 EUR; meanwhile, paracetamol cost EUR 48.56 and placebo cost EUR 2.77. In another study, the estimated mean cost per patient using glucosamine was USD 180, that of people using topical capsaicin was USD 236, and that of people using COX-2 NSAIDs such as celecoxib, cost nearly USD 500 per year [50]. Based on other research, standard treatment plus using glucosamine only cost about THB 150,000, but standard treatment plus etoricoxib cost up to approximately THB 420,000 [15]. These results show that the cost of using other medications is much higher than the cost of using glucosamine sulfate.

On the other hand, the study by Luksameesate et al. [15], 2022, in Thailand once again validated the intake of pCGS in the standard knee OA treatment, as it was regarded as a sufficient alternative to improve patients’ illnesses. As a result, the early initiation of pCGS would be less costly and more advantageous than delaying treatment. In addition, this publication also concluded that the combination of pCGS and etoricoxib in the treatment of knee OA is cost-effective at the willingness-to-pay threshold in Thailand, but the combination was only confirmed within the country [15]. WTP is based on the maximum price a customer/patient is willing to pay for a particular product or outcome; this range could be utilized to establish a generally accepted criterion [56]. The World Health Organization (WHO) recommends a threshold of one to three times the GDP per capita for the cost of investing in one disability-adjusted life year (DALY), which is widely known and frequently mentioned when considering CE standards. The former evaluation assumes that, if an intervention can yield one QALY per year for less than the GDP per capita, the subsequent value added will outweigh the cost of the investment [57,58]. Interventions with an ICER below the accepted ceiling threshold could then be considered cost-effective.

A study conducted in 2014 mentioned that utilizing glucosamine demonstrated therapeutic effects, along with reasonable expenses, but there have yet to be any studies that exclusively focus on the economic aspects or a cost-related examination of this intervention so far [24]. Additionally, another report in 2010 concluded that glucosamine sulfate was prioritized as a long-term analgesic because of its fair affordability and appropriate safety profile, as demonstrated by ICER analysis [41].

Furthermore, the clinical effects of pCGS in controlling OA symptoms have been demonstrated to be remarkably effective for improving pain and functional impairment [39]. Previously, a 2012 review of 78 outpatients with knee OA revealed that the administration of glucosamine sulfate was less money-consuming and had greater efficacy than the administration of glucosamine hydrochloride [59]. Ultimately, it can be said that the role and cost-effectiveness of glucosamine sulfate in the treatment of osteoarthritis have been further emphasized and defined.

There is a relatively large difference in treatment costs between high-income countries and low/middle-income countries, including the cost of glucosamine. Moreover, the population in high-income countries is often an aging population, which causes a large burden of disease for such countries. As people’s life expectancy is increasing, the burden of disease is becoming increasingly serious. Each country’s financial system must be adequately equipped to cope with this situation. In addition, countries need more supportive policies for the elderly and the poor to have access to health services.

However, we must admit there were limitations in our research. Firstly, in our systematic review, the number of studies that we collected was insufficient to fully support the cost-effectiveness of glucosamine. Secondly, since some studies were published a long time ago, their cost relevance is weak in today’s context. Lastly, the studies were conducted in the UK, Thailand, and some European countries, such as Germany and Spain. Thus, data evaluating the cost-effectiveness of glucosamine in Asia, especially in Vietnam, are limited.

## 5. Conclusions

pCGS is cost-effective in the treatment of mild and moderate osteoarthritis. In more severe OA, pCGS was deemed not to be cost-effective. We observed that patented different formulations of glucosamine were more cost-effective than OFG. A QALY gain was seen when glucosamine was administered in addition to conventional treatment.

## 6. Future Directions

The selected studies in this review show that the cost-effectiveness of glucosamine use has been demonstrated in the short-term (most of the studies evaluated the cost-effectiveness of glucosamine use lasted from about 6 months to 1 year). However, since osteoarthritis is a chronic disease, patients with osteoarthritis will need to take medication for most of their lives if no alternative therapy, such as joint replacement, is available. Therefore, future research should focus on performing long-term studies on the costs and health outcomes of glucosamine to assess its effectiveness more accurately. Moreover, further research is still needed to evaluate the actual use, effectiveness, adverse events, and cost-effectiveness of glucosamine use in Vietnam and other countries to provide evidence to support policies.

## Figures and Tables

**Figure 1 healthcare-11-02340-f001:**
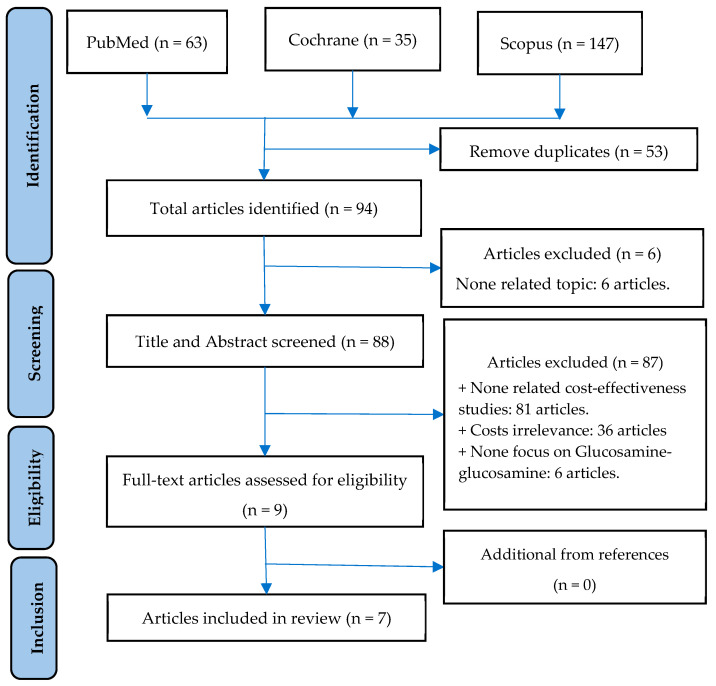
Process of article selection.

**Table 1 healthcare-11-02340-t001:** Summary of articles’ information.

		Number of Articles
Drugs used	Glucosamine and OTC drugs (NSAIDs, paracetamol)	3
	Glucosamine	4
Type of glucosamine	pCGS	3
	pCGS and OFG	4
OA site	Knee	5
	All	2

**Table 2 healthcare-11-02340-t002:** (**a**). Characteristics of selected studies. (**b**). Characteristics of selected studies.

**(a)**							
**No.**	**Study, Year, and Country**	**Subjects**	**Intervention**	**Perspective**	**Method**	**Time Horizon**	**Costs of Glucosamine**
1	Bruyère et al. [46], 2023, Thailand	OA patients	pCGS vs. OFG vs. placebo	Healthcare	CEA	-	USD 27.78/powder pCGS, USD 27.22/tablet pCGS. USD 14.61/powder OFG, USD 10.80/tablet OFG.
2	Luksameesate et al. [15], 2022, Thailand	Patients ≥ 45 years old with mild-to-moderate pain and no comorbidities	pCGS combined with etoricoxib vs. glucosamine monotherapy	Societal	CEA	Lifetime	-
3	Bruyère et al. [47], 2021, Germany	OA patients >40 years old	pCGS vs. OFG	Healthcare	CEA	-	-
4	Bruyère et al. [48], 2019,	OA patients >40 years old	pCGS vs. OFG	Healthcare	CEA	-	0.9 EUR/day for pCGS, 0.55 EUR/day for OFG
5	Scholtissen et al. [41], 2010 Spain, Portugal	Knee OA patients with average age of 63 years old	GS vs paracetamol vs. placebo	Healthcare	CEA	6 months	-
6	Black et al. [49], 2009, UK	Knee OA patients	GS/GH vs. chondroitin sulfate vs. GS and chondroitin	National healthcare system	CEA	Lifetime	£221 (1-year)
7	Segal et al. [50], 2004, Australia	OA patients	Interventions for arthritis, including glucosamine	National healthcare system	CUA	-	USD 180 (1-year)
**(b)**							
**No.**	**Study, Year, and Country**	**Subjects**	**Intervention**	**Model Type**	**Duration**	**Sensitivity Analysis**	**Discount** **Rate**
1	Bruyère et al. [46], 2023, Thailand	OA patients	pCGS vs. OFG vs. placebo	Grootendorst model	6 months	-	-
2	Luksameesate et al. [15], 2022, Thailand	Patients ≥ 45 years old with mild-to-moderate pain and no comorbidities	pCGS combined with etoricoxib vs. glucosamine monotherapy	Markov model	6 months	One-way; PSA	3%
3	Bruyère et al. [47], 2021, Germany	OA patients >40 years old	pCGS vs. OFG	Grootendorst model	3 years	-	-
4	Bruyère et al. [48], 2019,	OA patients >40 years old	pCGS vs. OFG	Grootendorst model	3 years	One-way	-
5	Scholtissen et al. [41], 2010 Spain, Portugal	Knee OA patients with average age of 63 years old	GS vs. paracetamol vs. placebo	Mathematical—decision model	6 months	PSA	-
6	Black et al. [49], 2009, UK	OA patients	Interventions for OA including glucosamine	Mathematical—decision model	1 year	-	5%
7	Segal et al. [50], 2004, Australia	Knee OA patients	GS Sulfate/hydrochloride vs. chondroitin sulfate vs. GS and chondroitin	Cohort model	1 year	One-way	3.5%

OA: osteoarthritis; pCGS: crystalline glucosamine sulfate; OFG: other formulations of glucosamine; PSA: probabilistic sensitivity analysis.

**Table 3 healthcare-11-02340-t003:** Cost-effectiveness-related data.

No.	Study, Year, and Country	Comparator	Cost	QALY Gain	ICER	Conclusion
1	Bruyère et al. [46], 2023, Thailand	pCGS vs. OFG	**At 3 months** pCGS: USD 53.805 OFG: USD 100.44 **At 6 months** pCGS: USD 126.1359	**At 3 months** pCGS: 0.017 OFG: 0.0031 **At 6 months** pCGS: 0.0411 OFG: 0.0048	**At 3 months** pCGS/PBO: 3165 USD/QALY OFG/PBO: 32,400 USD/QALY **At 6 months** pCGS/PBO: 3069 USD/QALY OFG/PBO: placebo better	pCGS is cost-effective at threshold of 3260 USD/QALY pCGS is more cost-effective than OFG
2	Luksameesate et al. [15], 2022, Thailand	pCGS + standard care vs. standard care	-	0.87	Dominant	Early addition of pCGS into standard care treatment early is cost-saving and more effective compared with standard care alone
3	Bruyère et al. [47], 2021, Germany	pCGS vs. OFG	**At 3 months** pCGS: EUR 77.0964 OFG: EUR 208.854 **At 6 months** pCGS: EUR 183.0003 **At 36 months** pCGS: EUR 2785.2712	**At 3 months** pCGS: 0.0164 OFG: 0.0036 **At 6 months** pCGS: 0.0413 OFG: 0.0044 **At 36 months** pCGS: 0.2701	**At 3 months** pCGS/PBO: 4701 EUR/QALY OFG/PBO: 58,015 EUR/QALY **At 6 months** pCGS/PBO: 4431 EUR/QALY OFG/PBO: Placebo better **At 36 months** pCGS/PBO: 10,312 EUR/QALY	pCGS is more cost-effective than OFG
4	Bruyère et al. [48], 2019	pCGS vs. OFG	**At 3 months** pCGS: EUR 90.234 OFG: EUR 151.009 **At 6 months** pCGS: EUR 209.413 **At 36 months** pCGS: EUR 3162.910	**At 3 months** pCGS: 0.0169 OFG: 0.00303 **At 6 months** pCGS: 0.0435 OFG: 0.00424 **At 36 months** pCGS: 0.2742	**At 3 months** pCGS/PBO: 5347.2 EUR/QALY OFG/PBO: 49,737.4 EUR/QALY **At 6 months** pCGS/PBO: 4807.2 EUR/QALY OFG/PBO: Placebo better **At 36 months** pCGS/PBO: 11,535.5 EUR/QALY	pCGS is more cost-effective than OFG
5	Scholtissen et al. [41], 2010 Spain, Portugal	GS vs. paracetamol, GS vs. placebo	-	-	GS/paracetamol: −1376 EUR/QALY GS/placebo: 3617.47 EUR/QALY	GS is highly cost-effective vs. paracetamol
6	Black et al. [49], 2009, UK	GS adding conventional vs. conventional care	GBP 2346.85	0.11	21,335 GBP/QALY	Addition of GS therapy to current care is cost-effective at threshold of 22,000 GBP/QALY
7	Segal et al. [50], 2004, Australia	GS vs. NSAIDs	USD 180.024	0.052	3462 USD/QALY	Glucosamine is cost-effective

GS: glucosamine; pCGS: crystalline glucosamine sulfate; OFG: other formulations of glucosamine; NSAIDs: non-steroidal anti-inflammatory drugs.

## Data Availability

Data were collected from PubMed, Scopus, and Cochrane databases.

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
