# Peer review of "Cost-Effectiveness of Glucosamine in Osteoarthritis Treatment: A Systematic Review"

_healthcare, 2023, doi:10.3390/healthcare11162340_

Round 1
Reviewer 1 Report
· Statements through lines 65-68 should be referenced.
· The manuscript does not provide justification for selection of 3 specific data bases for search. Why did the study not consider others such as Web of Science, Google Scholar, etc?
· Are there any differences in findings for low/middle income vs. high income countries?
· The manuscript should provide a discussion on potential differences in healthcare financing system and measurement of QALYs in the countries included in the study.
· Policy implications of findings should be discussed in detail.
· Conclusion section of the manuscript is weak and should be improved.
A proofread would be useful.
Author Response
Dear Reviewer 01,
Thank you for your support. Please see the attachment.
Best regards,

Reviewer 2 Report
Dear Authors,
This manuscript sheds insights on an ever-growing problem. However, the English needs substantial updating. I have highlighted &/ crossed out with the corrections and comments on the left side in comment boxes. I find the use of some of the rather old citations meaningless in the context of cost-effectiveness or the cost of treating a disease in today's landscape.
You cite the use of databases until July 2023. My comments is that I cannot imagine how is it possible to have reviewed the databases until July 2023 when I am reviewing this manuscript on the 1st August? Surely your last searches, selection, data extraction and writing the manuscript cannot have been completed by 30th June.
Please correct all glucosamine that are mid-sentence to being written in lower case, as is the norm for generic products. Oddly, you have been inconsistent in the way you have written it - sometimes upper case, others lower case. Similarly, crystalline glucosamine sulfate etc, unless they are registered tradenames, in which case you would need to include the trademark sign.
Please check the spelling of all ageing (not aging), I have highlighted some.
I think that you need far more recent citations for relevant data in the Introduction. For example:
The direct costs of osteoarthritis treatment can be as high as 1–2.5% of Gross National Product in countries such as the US, UK, Canada and Australia (27). That citation is from 1997, I don't think it has any bearing on the current impact on GDP / GNP.
Your introduction lacks chronology. For example, you reported Salmon et al, 2016 before reporting Torio et al, 2013. Therefore, from the reader's perspective, the timelines jump around and the costs become more demanding to compare. Since OA is increasing, cost of treatment is also increasing, and by using the chronology of publication dates, it adds value to the message. You bounce around so much that the reader loses the will to read on. USA in 2013, then Australia in 2015, then France in 2001 and then you have placed the indirect costs for Spain, UK and USA in one sentence (refs. 35 and 36) but these relate to studies conducted at different times.
Use of the QHES is not common in health economic studies - the most commonly used and validated tool is the Oxford Levels of Evidence. https://www.cebm.ox.ac.uk/resources/levels-of-evidence/ocebm-levels-of-evidence
The "Study selection process" section is poorly described, AND repeated. Please see the suggestion I have made, to which you can add how differences of opinion were managed, but please do so in the past tense. not "where there is any disagreement, the teams will conduct...", the study is completed, please report it in the past tense. Or remove that section about disagreement and include your study protocol as an appendix or in the supplementary information. The people who are likely to read this publication know how to manage literature searches.
Table 1 is confusing. I suggest that you tabulate it in a clearer way.
The mixture of currencies in section 3.4 and the lack of reference to the threshold cost/QALY and the willingness-to-pay threshold makes this section very jumbled.
You sometimes use paracetamol, and other times acetaminophen. I suggest that you use both, for example: acetaminophen (paracetamol) each time to help readers from different markets.
I don't agree that a systematic literature review and presenting the results offers a framework (Line 376, Page 15). Proposing a framework would entail doing an analysis of existing frameworks, identifying their strengths and weaknesses on which to develop a new and robust framework. Apologies, but you have not done this.

The English needs a major overhaul. I have highlighted and added suggestions in comment boxes, but working in PDF is a real nuisance, it would be easier in a Word doc with track changes.
Author Response
Dear Reviewer 02,
Thank you for your support. Please see the attachment.
Best regards,

Reviewer 3 Report
Dear Authors,
I congratulate you on your efforts conducting this extensive review and putting forward a nicely written report. Agreeably, osteoarthritis’ prevalence is increasing across regions worldwide, and does poses significant global health challenges including economic consequences. It is quite timely to evaluate the economic relevance of osteoarthritis treatment using the glucosamine option. The study design and its approach are appropriate. I particularly appreciate that the authors included detailed criteria used to include and exclude articles. Including a PRISMA flowchart is equally nice. With the obvious limitations due to the number and non-diverse number of articles included, the scope of the study objectives, and the wide exclusion criteria, the authors rightly noted areas for future research. Having a prospective study that assesses the effectiveness of glucosamine use, in relation to cost and health implication, would possibly give more insight into the subject matter.
I have included some comments and suggestions for your consideration.
Method: To enhance validity and reproducibility of the findings, consider including the search words and/or mesh terms used for each of the search database used for this work. Although some search terms are listed in lines 149 - 151, consider mentioning what applies to PubMed, Scopus, and Cochrane individually.
Result:
Line 186: The first sentence is a repetition of line 137. Consider avoiding a repeat of the date the selection process was conducted.
Consider including the initials of authors that independently assessed the articles (if this process was done).
Lines 227 - 228: Including the words "with" and "were" is inappropriate. Consider replacing the sentence with either of the following: "Our subjects included patients diagnosed with Osteoarthritis; two of the studies were knee-focused." or "Our subjects included patients diagnosed with Osteoarthritis, with two of the studies knee-focused."
Line 386: Consider replacing the word "system" with "systematic."
Lines 388-389: these sentences are not clear. Please provide more context with explanation to enhance easy comprehension by our diverse audience of readers.
Future Directions:
While I appreciate the call for further research in Vietnam, I am not sure I found how the study is tailored towards Vietnam. Consider generalizing the call for further research, as a means of closing the non-diversity of available articles gaps.
While I commend the due diligence of the authors putting together a comprehensible piece, I encourage them to conduct a thorough grammar check. For example:
Lines 227 - 228: Including the words "with" and "were" is inappropriate. Consider replacing the sentence with either of the following: "Our subjects included patients diagnosed with Osteoarthritis; two of the studies were knee-focused." or "Our subjects included patients diagnosed with Osteoarthritis, with two of the studies knee-focused."
Line 386: Consider replacing the word "system" with "systematic"
Lines 388-389: these sentences are not clear. Please provide more context with explanation to enhance easy comprehension by our diverse audience of readers.
Author Response
Dear Reviewer 03,
Thank you for your support. Please see the attachment.
Best regards,

Round 2
Reviewer 1 Report
The current version may be published.
Author Response
Dear Reviewer 01,
Thank you for all your support and help to make this manuscript better.
Best regards,
Reviewer 2 Report
Dear Authors,
Thanks, it's taking shape! I really appreciate how hard it is to write well in English when it is your second or third language. But even native English speakers make many grammatical errors!
Please address the comments I've made in this updated version.
I strongly recommend that you review your use of the term ICER, especially in Line 234. Firstly, having already introduced the abbreviation, there is no need to do so again, and especially NOT incorrectly, since ICER = Incremental cost-effectiveness ratio and NOT increased cost-effectiveness ratio. Furthermore, the ICER is not a measure of disease burden. The ICER is the economic value of a new intervention versus another. Disease burden is the impact of a health problem or a disease as measured by the financial burden, the physical burden, emotional burden, morbidity, mortality, and is often expressed as the QALY.
All the best!

The manuscript needs a few minor corrections
Author Response
Dear Reviewer 02
Thank you for all your support and help to make this manuscript perfect. All your contributions are greatly appreciated. Your comments have been edited in the revised version.
Best regards,
